# Analyzing Intra-Cycle Velocity Profile and Trunk Inclination during Wheelchair Racing Propulsion

**DOI:** 10.3390/s23010058

**Published:** 2022-12-21

**Authors:** Yoann Poulet, Florian Brassart, Emeline Simonetti, Hélène Pillet, Arnaud Faupin, Christophe Sauret

**Affiliations:** 1Centre d’Études et de Recherche sur l’Appareillage des Handicapés, Institution Nationale des Invalides, 75007 Paris, France; 2Arts et Métiers Institute of Technology, Université Sorbonne Paris Nord, IBHGC—Institut de Biomécanique Humaine Georges Charpak, HESAM Université, 151 Bd de l’Hôpital, 75013 Paris, France; 3Laboratoire IAPS, Université de Toulon, 83130 Toulon, France

**Keywords:** wheelchair sports, propulsion, racing, intra-push, velocity profile, kinematics

## Abstract

The analysis of intra-cycle velocity profile of manual wheelchair (MWC) users has been used to highlight the significant role of trunk inertia in propulsion biomechanics. Maximal wheelchair linear velocity has previously been observed to be reached after the release of the handrims both during sports activities and daily life propulsion. This paper provides a combined analysis of linear velocity and trunk kinematics in elite wheelchair racing athletes during straight-line propulsion at stabilized speeds. MWC and trunk kinematics of eight athletes (level: 7 elite, 1 intermediate; classification: T54 (5), T53 (2) and T52 (1)) were monitored during 400 m races using inertial measurement units. An average propulsion cycle was computed for each athlete. The main finding of this article is the difference in propulsion patterns among the athletes, exhibiting either 1, 2 or 3 peaks in their velocity profile. A second peak in velocity is usually assumed to be caused by the inertia of the trunk. However, the presence of a second velocity peak among more severely impaired athletes with little to no trunk motion can either be associated to the inertia of the athletes’ arms or to their propulsion technique.

## 1. Introduction

Performance in WheelChair Racing (WCR) has greatly improved since the emergence of this sport in the 1950s [1] due to its professionalization and to the technological improvements in wheelchair design and manufacturing [2]. However, this trend has slowed drastically over the past decade to the point that no significant improvement in performance has been observed between 2009 and 2018 [3]. Performance in WCR can be defined as the time required to complete a given distance, making the evolution of manual wheelchair (MWC) speed during the race the main outcome parameter to study. At first glance, MWC velocity appears to be solely due to the athlete’s mechanical action on the handrims. Thus, propulsion technique, defined as the biomechanical strategy used by athletes during a stroke, and which differs from athlete to athlete due to their different impairments, is one of the major factors influencing WCR performance. Since athletes present different degree of activity limitation due to their various impairments, athletes with similar capacities are grouped together for competition through a process called classification. To better understand WCR propulsion and to increase its performance, several researchers studied the evolution of MWC velocity [4,5] or the impact of propulsion technique on WCR performance [6]. Due to the ease of instrumentation and in order to increase repeatability, the first studies were carried out in the lab on a wheelchair roller ergometer.

Since then, multiple studies have been carried out overground using adapted instrumentation, allowing to highlight the importance of the upper body on MWC velocity during propulsion [7]. For instance, in a case study involving an international level wheelchair racer during a 10m sprint, Moss and colleagues observed that for each propulsion cycle, maximal velocity was achieved after the release of the handrim, during the extension of the trunk [8]. Achieving maximal velocity after the release of the handrim was contradictory with the traditional description of wheelchair propulsion as being divided into a “propulsive” phase when the hands of the athlete are in contact with the handrim followed by a “recovery” phase after the release of the handrim [9]. This study combined the use of a velocometer [10] to record wheelchair velocity and a motion capture system to monitor head, arm, and trunk kinematics. Subsequent studies obtained comparable results both in WCR athletes, using a speed radar [11], and in able-bodied volunteers using an instrumented MWC [12].

Reaching maximal velocity after the release of the handrim reveals the existence of a second source of forward driving force, independent from the handrims. This second source was hypothesized to be the action of the upper body on the seat [8,12]. This phenomenon can only be observed by using data acquisition systems with a sufficiently high sampling rate under ecological propulsion conditions. Indeed, a roller ergometer only allows the wheel to be accelerated by the actions applied by the athlete on the handrims, preventing the described phenomenon from occurring. Reaching maximal velocity after the release of the handrim reveals the existence of a second source of forward driving force, independent from the handrims. This second source was hypothesized to be the action of the upper body on the seat [8,12]. Such a phenomenon could not have been observed with under-sampled acquisition systems (e.g., video cameras, timing gates etc.) or relying solely on a roller ergometer which, by its mechanical design, only allows the wheel to be accelerated from the actions applied by the athlete on the handrims, preventing the described phenomenon from occurring.

Over the past decade, technological advancements have made Inertial Measurement Units (IMUs) small and wireless. These wearable sensors, consisting of a magnetometer, a gyroscope and an accelerometer, allow the determination of the sensor orientation, rotational velocity and acceleration. Through the last twenty years, they have become accessible to research teams around the world, making “out-of-lab” or “in the field” experiments feasible and providing more reliable results. Pansiot and colleagues [13] developed a methodology for wheelchair displacement and orientation monitoring using IMUs placed on both rear wheels of a wheelchair. Speed measurement and analysis using IMUs in wheelchair racing was extensively discussed by Franz Fuss [14]. Using such sensors, multiple studies on both WCR [15,16] and MWC court sports (tennis, basketball, and rugby) were then carried out [17]. In particular, while studying sprint among elite female wheelchair basketball athletes, Brassart and colleagues found the range of motion of the trunk in flexion/extension to be positively correlated to MWC mean acceleration per push phase while being uncorrelated to its mean deceleration per recovery phase [18].

However, the use of IMUs to accurately monitor MWC velocity provides complex data containing acceleration, angular velocity, and orientation at high sampling rate (i.e., usually 60 to 120 Hz). Analyzing MWC velocity in depth, a preliminary study presented as a conference paper [19] found that each of the four studied athletes exhibited a unique velocity profile. Based on this preliminary study, the objective of the present paper was to analyze MWC and trunk kinematic data (i.e., MWC linear velocity, trunk inclination, trunk inclination speed) and interpret them regarding impairment and WCR level over a larger number of athletes. For this purpose, seven elite and one intermediate WCR athletes with different classification levels, and thus different trunk muscle control, were examined over 400 m races. We hypothesized that the athletes would exhibit different intra-cycle velocity profile and trunk inclination pattern, with a clear distinction between athletes of different classifications.

## 2. Materials and Methods

This paper merges data from two similar but distinct studies which both obtained ethical agreement beforehand (n°IRB00012476-2021-05-02-84 and N°IDRCB: 2020-A02919-30). The differences in methodologies between both studies will be discussed in the following paragraphs.

In total, eight athletes (seven elite, i.e., U23 and senior international and paralympic level, and one intermediate level) gave informed consent to participate. Their demographics (gender and age), classification as defined by the International Paralympic Committee, and wheelchair configurations when available (camber, rear wheel and handrim diameter, wheelbase, position of the center of gravity and mass moment of inertia of the wheelchair around the vertical axis or “yaw mass moment of inertia”) are provided in Table 1. Classification determines which athletes are eligible to compete together based on the degree of activity limitation resulting from their impairment. Thus, in para-athletics, T54 athletes display partial to normal trunk control, whereas T53 and T52 athletes do not have abdominal or lower spinal muscles activity. Additionally, T52 athletes present low level arm and shoulder impairment.

### 2.1. Experimental Protocol

The athletes were asked to perform a high-intensity 400 m race on athletics tracks. The acquisition recording started from standstill with the athlete in his starting position and ended when the wheelchair came to a complete stop after the race and a rest lap if needed.

Athletes and their wheelchairs were equipped with three wireless IMUs (either Xsens, Netherlands, 100 Hz or WheelPerf System, AtoutNovation, France, 128 Hz), one placed on the inside of each rear wheel using strong double-sided adhesive tape (Figure 1a), and one on the athlete’s trunk attached to a Velcro strap around the athlete’s trunk, preferably on his sternum otherwise on his back between the shoulder blades to prevent discomfort and displacement of the IMU during propulsion. Using wireless IMUs allowed the athletes to propel their wheelchairs unhindered by the experimental setup but required a permanent Bluetooth connection to a computer within a 20 m range, requiring an experimenter to follow the athlete on a bicycle. The difference in frequency of acquisition between the two brands of IMUs has been addressed by down sampling the 128 Hz signals of the WheelPerf sensor to 100 Hz using MATLAB in order to correspond to the sampling frequency of the Xsens sensor. The same data processing pipeline was then used for both IMUs data.

### 2.2. Data Processing

#### 2.2.1. True Wheel Velocity

Because MWC wheels can be considered to roll without slipping in WCR, MWC linear velocity, MWC linear velocity ($V_{x}$), can be computed by multiplying true wheels rotational velocities around their axis of rotation (${\dot{\beta }}_{right\_wheel}$ and ${\dot{\beta }}_{left\_wheel}$) and wheels radius (r), as expressed in Equation (1) [14]:(1)$$V_{x}=r\left(\frac{{\dot{\beta }}_{left\_wheel}+{\dot{\beta }}_{right\_wheel}}{2}\right)$$

Since sports wheelchairs use positive camber (α) for stability and maneuverability purposes, gyroscopes placed on the wheels capture both true wheel rotational velocity ($\dot{\beta }$) and the rotational velocity of the wheelchair around the vertical axis ($\dot{\theta }$). The methodology proposed by Pansiot [13] decouples these two rotational terms to identify true wheel rotational velocity (Equations (2)–(9)). Defining the MWC frame with *x*-axis antero-posterior, *y*-axis vertical and *z*-axis mediolateral to the right (Figure 1a), wheels angular velocities (${\vec{\Omega }}_{Wheel/R0}=\Omega_{x},\;\Omega_{y},\Omega_{z}$) are composed as follows:(2)$${\vec{\Omega }}_{wheel/R0}={\vec{\Omega }}_{wheel/MWC}+{\vec{\Omega }}_{MWC/R0}$$
(3)$${\vec{\Omega }}_{wheel/R0}=\dot{\beta }\vec{z_{wheel}}+\dot{\theta }\vec{y_{MWC}}$$

The following rotation matrix is used to express the last term in the wheel frame:(4)$$R_{MWC\to wheel}=R_{z}\left(\beta \right)R_{x^{\prime }}\left(\alpha \right)$$
(5)$$\left(\beta \right)=\left[\begin{array}{ccc} \cos\beta & \sin\beta & 0\\ -\sin\beta & \cos\beta & 0\\ 0 & 0 & 1 \end{array}\right]\;\;\;R_{x\prime }\left(\alpha \right)=\left[\begin{array}{ccc} 1 & 0 & 0\\ 0 & \cos\alpha & -\sin\alpha \\ 0 & \sin\alpha & \cos\alpha \end{array}\right]$$

Solving for Equation (3):(6)$${\left(\begin{array}{c} \Omega_{x}\\ \Omega_{y}\\ \Omega_{z} \end{array}\right)}_{Wheel}={\left(\begin{array}{c} \dot{\theta }\cos\alpha \sin\beta \\ \dot{\theta }\cos\alpha \cos\beta \;\\ \dot{\beta }-\dot{\theta }\sin\alpha \end{array}\right)}_{Wheel}$$

From Equation (6) derives:(7)$$\Omega_{x}^{2}+\Omega_{y}^{2}={\left(\dot{\theta }\cos\alpha \right)}^{2}$$

Thus, the angular velocity of the wheelchair around the vertical axis can be expressed as:(8)$$\dot{\theta }=\pm \sqrt{\Omega_{x}^{2}+\Omega_{y}^{2}\;\;\;}\cos^{-1}\alpha$$

Substituting into Equation (6), the true wheel rotational velocities can be obtained:(9)$$\dot{\beta }=\Omega_{z}\pm \sqrt{\Omega_{x}^{2}+\Omega_{y}^{2}\;\;\;}\;\;\tan\alpha$$

In Equations (8) and (9), minus signs are used for the right wheel and plus signs for the left wheel.

#### 2.2.2. IMU Realignment

Additionally, because the IMUs were placed on the wheel spokes, the main axes of the IMUs cannot be considered to be aligned with the wheel axis (Figure 1b). Thus, the IMUs positioned on the wheels do not measure wheel angular velocity presented in the wheel reference frame. To take this misalignment into account, short straight-line displacements were recorded. For each wheel, the rotation matrix distributing the measured rotational velocity (${\left[Gyr\right]}_{IMU}=\left(Gyr_{x},Gyr_{y},Gyr_{z}\right)$) around a new axis aligned with the wheel axis was computed. This rotation matrix $R_{IMU\to wheel}$ intervenes as follows:(10)$${\vec{\Omega }}_{wheel/R0}={\left[Gyr\right]}_{wheel}=R_{IMU\to wheel}\times {\left[Gyr\right]}_{IMU}$$

This rotation matrix is the combination of consecutive rotations, ε and γ, around the *x* axis of the wheel then around the newly obtained *y*’ axis (Equation (10)).
(11)$${R_{IMU\to wheel}}^{-1}=R_{wheel\to IMU}=R_{y\prime }R_{x}$$
(12)$$R_{x}=\left[\begin{array}{ccc} 1 & 0 & 0\\ 0 & \cos\varepsilon & -\sin\varepsilon \\ 0 & \sin\varepsilon & \cos\varepsilon \end{array}\right]\;\;\;R_{y\prime }=\left[\begin{array}{ccc} \cos\gamma & 0 & \sin\gamma \\ 0 & 1 & 0\\ -\sin\gamma & 0 & \cos\gamma \end{array}\right]$$

Thus, the rotation matrix can be written as follows:(13)$$R_{wheel\to IMU}=\left[\begin{array}{ccc} \cos\gamma & \sin\gamma \sin\varepsilon & \sin\gamma \cos\varepsilon \\ 0 & \cos\varepsilon & -\sin\varepsilon \\ -\sin\gamma & \cos\gamma \sin\varepsilon & \cos\gamma \cos\varepsilon \end{array}\right]$$

By dividing the gyroscope data by their norm (∥Gyr∥), Equation (10) becomes:(14)$$\frac{Gyr_{IMU}}{∥Gyr∥}=R_{wheel\to IMU}\times \left[\begin{array}{c} 0\\ 0\\ 1 \end{array}\right]$$

That means:(15)Gyrx∥Gyr∥Gyry∥Gyr∥Gyrz∥Gyr∥=sinγcosε−sinεcosγcosε

Finally, Equation (15) allows the identification of ε and γ:(16)$$\gamma =\tan^{-1}\left(\frac{Gyr_{x}}{Gyr_{z}}\right)\;\varepsilon =\tan^{-1}\left(\frac{-Gyr_{y}\times \cos\gamma }{Gyr_{z}}\right)$$

By feeding these two angles back into the rotation matrix and then solving for Equation (10), the data measured by the gyroscopes of the IMUs can be expressed in a frame with the *z*-axis aligned with the wheel axis of rotation, thus representing wheel rotational velocity (${\vec{\Omega }}_{wheel/R0})$.

#### 2.2.3. Trunk Kinematics

Trunk kinematics were monitored to quantify the impact of the trunk on MWC velocity. For this purpose, trunk inclination speed was measured using the gyroscope of the IMU placed on the torso of each athlete. Sensor-to-segment calibration was performed using a combination of methodologies [20]. For the alignment of the mediolateral axis of the IMU with the medio-lateral axis of the athlete, a similar approach to the one presented in Section 2.2.2 was performed. To do so, the frame with the most trunk flexion speed during the first straight-line propulsion cycle was used. Then, trunk flexion angle was computed from the rotation matrices provided by the Xsens IMUs and from the quaternions provided by the WheelPerf IMUs, both following an y-z-x sequence. Because two types of IMUs were used, two methodologies had to be developed. Indeed, on the one hand, for subjects equipped with Xsens IMUs (see Table 1), Xsens’s orientation calibration tool “heading reset” was used with all IMUs aligned on the flat ground to reset and synchronize IMUs’ reference frames. This allowed the computed Euler angles to be used directly, knowing that an angle of 0° corresponded to an IMU parallel to the ground. On the other hand, athletes equipped with the WheelPerf System (E, F and H) were asked to stand still in their starting position before the race. In this case, inclination angle of the IMU was visually assessed by the experimenter (30° for E and 0° for F and H). Then, the measured offset was subtracted to the computed Euler angles. Since this method is prone to error, the data of the three athletes processed this way will be identified in the rest of the paper.

In this paper, trunk flexion angle will be considered to be 0° for an athlete leaning forward in a horizontal position and 90° for an athlete sitting upright in his MWC.

#### 2.2.4. Cycle Normalization

For each athlete, propulsion cycles were manually identified based on visual inspection of the repeated pattern on the wheel rotational velocity signal (Figure 2). Propulsion cycles that occurred outside straight lines (i.e., when rotational velocity from the right and left wheel differed) or during start-up (before maximal speed was reached) were discarded in order to gather only cycles of straight-line propulsion at steady state. Cycles were then normalized from 0% to 100% of propulsion cycle. Finally, for each athlete an average cycle with its standard deviation corridor was defined.

The previously described outcome parameters (MWC linear speed, trunk inclination, trunk inclination speed) as well as cycle time were computed for each cycle then averaged per subject to characterize the average propulsion cycle of each athlete.

#### 2.2.5. Data Analysis

In order to study the interactions between the different measured kinematic and spatiotemporal parameters defining wheelchair propulsion, Pearson correlation coefficients (R) were computed between each parameter using MATLAB. For this purpose, classification level was quantified as 1 for T52, 2 for T53 and 3 for T54. Considering the number of velocity peaks, clear and distinct peaks were given a weight of 1 when smaller peaks were only given a weight of 0.5. Accordingly with previous literature, correlation coefficients were interpreted as follows: absence of or little correlation (R < 0.25), weak (0.25 ≤ R < 0.50), moderate (0.50 ≤ R < 0.70), strong (0.70 ≤ R < 0.90) and very strong (R ≥ 0.90) [21].

## 3. Results

On average, 34 cycles per athletes were considered (Table 2). The experimental setup resulted in having only 12 exploitable propulsion cycles for athlete C. However, because propulsion cycles of such high-level athletes are extremely repeatable, 12 cycles were considered to be enough to include athlete C in the study. Outcome parameters were computed for each cycle of each athlete. Averages and standard deviations were then computed for each athlete.

### 3.1. MWC Linear Velocity

Mean propulsion cycle speed varied between 5.6 m/s for athlete G of intermediate level to 8.56 m/s for athlete F. However, in order to highlight the differences in velocity variation, the average propulsion cycle of each athlete was centered on their average cycle speed (Figure 3, first row). The primary results obtained by looking at the curves is the multiplicity of velocity patterns exhibited by the eight athletes as well as the repeatability of each athlete illustrated by the narrow, shaded corridor representing the mean value plus and minus one standard deviation. Even among athletes of the same classification level, variations in velocity pattern were observed: T54 athletes with complete trunk control showed patterns with either one sharp velocity peak (B and G), two clearly distinct peaks (A) or two smaller peaks almost creating a plateau (F and H). Athletes with lower trunk control either presented a two-peak velocity profile (D and E) with less amplitude than athlete with higher trunk control (A) or a profile with a third peak (C).

The absolute linear velocity variation, which is the difference between the maximal and minimal instantaneous velocities during each athlete’s average propulsion cycle was comprised between 0.48 m/s and 0.60 m/s, except for athlete B who presented a notably higher variation of 0.83 m/s.

### 3.2. Trunk Kinematics

Subjects with a T54 classification level (A, B, F, G and H) exhibited much more average trunk flexion amplitude per propulsion cycle (24.8°) than athletes of classification T53 and T52 with low to no trunk control (11°). Athlete D presented no variation in his trunk inclination. 

Average trunk flexion was comprised between 0° and 14° for six out of eight participants. Indeed, athlete G of intermediate level was sitting in a more upright position and exhibited an average trunk flexion of 44° and athlete A an average trunk flexion of 25°.

Trunk flexion speed was difficult to correlate with other results. Except for athlete F presenting a higher variability in this parameter, all the other athletes were once again very repeatable. 

Additionally, an interesting result deriving from trunk kinematics is the observation of multiple peaks in the velocity profile of T53 and T52 athletes (especially D and E) despite their low trunk flexion amplitude (respectively 3° and 9°). 

### 3.3. Correlations

Pearson correlation coefficients (R) between all variables are presented in Table 3. As expected, average velocity and time to perform the 400m race were found to be very strongly negatively correlated (R = −0.94, *p* = 0.0005). Naturally, average trunk flexion angle was very strongly positively correlated to both minimal and maximal trunk flexion angles (R = 0.94, *p* = 0.0006; R = 0.92, *p* = 0.001, respectively). Trunk flexion amplitude was found to be strongly correlated to velocity variation during propulsion (R = 0.82, *p* = 0.013). Finally, trunk flexion speed and trunk flexion amplitude were strongly correlated (R = 0.83, *p* = 0.010).

It is interesting to note that classification level, the number of peaks present in each athlete’s intra-cycle velocity profile and cycle time were found to be at most moderately correlated with any of the studied parameters, except for the previously highlighted correlation between the time to perform 400 m and average velocity.

## 4. Discussion

The objective of this study was to investigate whether trunk kinematics could explain the different velocity profiles exhibited by WCR athletes in light of their impairment level. To our knowledge, this study is the first to provide detailed information about WCR athletes intra-cycle velocity profile and trunk kinematics during overground propulsion. More importantly, it demonstrates the feasibility of in situ measurements using IMUs to study WCR propulsion kinematics.

### 4.1. Considerations on Data Processing

An important result derived from this study is that the automation of cycle identification using classical technique based on the search of local minima and maxima in MWC linear velocity is not possible in WCR athletes. This conclusion is the same based on trunk flexion angle and trunk flexion velocity. Indeed, looking at athlete D, it appears logical that trunk flexion and trunk flexion speed cannot be used for this purpose. Hence, other sensors, such as instrumented handrims measuring propulsion torque or pressure sensors placed on the gloves for example could be necessary for cycle identification and, moreover, for the distinction between push and recovery phases.

### 4.2. Propulsion Analysis

In line with the results obtained in our preliminary study [19], the velocity pattern of the eight athletes were found to be notably different. Only two athletes (B and G) exhibited a single peak in their velocity profile, whereas all the other athletes presented two to three peaks, more or less clearly visible. 

Visually, trunk flexion amplitude is a discriminating factor between athletes of different trunk control. Although only moderately correlated (R = 0.64), trunk flexion amplitude logically appears to be positively linked to classification level when looking at Figure 3. This relatively low correlation coefficient might result from the smaller number of T53 and T52 athletes and from the relatively high amplitude exhibited by athlete C despite his T53 classification. Indeed, the T53 classification level gathers athletes with partial to no trunk control.

Additionally, trunk flexion amplitude was positively correlated (R = 0.82, *p* = 0.013) to the magnitude of velocity variation during the propulsion cycle. Despite this strong correlation, it is interesting to note that athletes with lower trunk control (C, D and E) also exhibited multiple velocity peaks in their profiles. This result, particularly noticeable for subjects D and E, shows that trunk is not the only contributor to MWC velocity fluctuation during the propulsion cycle. This suggests that the arms linear and angular momentum of the athlete are high enough, which in turn means they are heavy enough and/or moving with enough acceleration, to impact the velocity of the entire {MWC + athlete} system during propulsion. This experimental result comforts previous observations made using a numerical simulation of WCR propulsion [22].

Regarding trunk flexion amplitude, despite being technically more impaired and classified T52, athlete E displayed a larger amplitude than athlete D. This could be due to his trunk being driven upward when exerting an action on the handrims. 

Despite being positively correlated to trunk flexion amplitude, trunk flexion speed remains difficult to interpret. It would have been beneficial to be able to assess the linear momentum of the trunk center of mass in the forward direction for a better analysis of the trunk contribution to the fluctuations of the MWC linear velocity. Note, the noisy aspect of trunk flexion speed of subject F might be due to an IMU that was not sufficiently secured on the athlete’s trunk or to a too large motion caused by back muscles contraction during the propulsion cycle. 

### 4.3. Limitations and Perspectives

Naturally, some limitations should be considered. Firstly, the use of IMUs with internal memory allowing data acquisition from a distance would ensure less loss of data during the races. Secondly, measuring trunk inclination angle using a single IMU provides results under the strong assumption that the trunk is a rigid body. Moreover, the exact alignment of the IMU with the trunk is unknown. Functional motions performed during the race were used for sensor-to-segment calibration in this study, but a preliminary functional calibration could help reducing the uncertainties on trunk inclination angle [20]. However, performing such a calibration would necessitate the athlete to maintain his trunk into a vertical position to use the acceleration of the gravity, which might not be feasible depending on the athlete’s trunk control and due to the high risk of tipping backwards in racing MWC, which are designed for a forward leaning position. A third limitation is the relatively small number of included athletes which weakens the statistical power of the article. Additionally, athletes performing wheelchair racing present various disabilities and strongly individual characteristics. Thus, Pearson correlation coefficients should be interpreted with care and the results of this paper considered as tendencies. Moreover, Pearson correlation coefficient only detect linear relationships between variables. If two variables were to form a non-linear relationship, this statistical test might not be appropriate. However, given the small number of high-level wheelchair racing athletes in the world, the recruitment of seven elite athletes to participate in such a study is already a step forward in para-athletics research. Finally, gathering the athletes by classification level rather than by disability allowed to investigate potential relationships in light of sport-specific functional limitations.

Although removing the limitations presented above would allow a more accurate analysis, we believe that these limitations do not interfere with the validity of the results and discussion presented in this paper. Thus, this paper succeeded in demonstrating the feasibility of using IMUs to monitor intra-cycle velocity profile and trunk inclination during wheelchair racing.

WCR athletes tend to lean forward on their wheelchair, with an average trunk flexion comprised between 0° (horizontal) and 14° for six out of the eight athletes studied. To further investigate the impact of trunk flexion on MWC kinematics, it would be interesting to study the horizontal acceleration of the trunk center of mass. This parameter would be representative of the impact of the trunk on MWC linear velocity because it would give more importance to trunk motion when the athlete is sitting in a more upright position, thus making the trunk center of mass oscillates in a more horizontal plane. Indeed, oscillating the trunk in an upright position would promote velocity fluctuation whereas oscillations around a horizontal position would mostly vary the load applied on the MWC wheels, which certainly affects rolling resistance but should have a much smaller impact on velocity. However, computing the horizontal acceleration of the trunk center of mass is challenging and first requires an accurate definition of the trunk segment with respect to the IMU fixed on the trunk (see above), and the estimation of the position of the trunk center of mass, which might be difficult depending on the athlete’s anthropology and impairment. Additionally, when looking at videos of the acquisitions, some athletes exhibited a “round back” during the races. Monitoring trunk inclination with a more adapted device allowing for the decomposition of the trunk into lumbar and thoracic segments could help interpreting the results.

## 5. Conclusions

Although there remain technical challenges to automate the study of in-situ intra-cycle velocity profile during overground manual wheelchair propulsion, this article provided evidence of the feasibility of the analysis of wheelchair racing kinematics using Inertial Measurement Units. Additionally, this study experimentally evidenced that trunk kinematics do not fully explain velocity variations after the release of the handrims in MWC propulsion. Further research in that line could provide more insight on the underlying mechanisms in play by using additional sensors and inertial models adapted to WCR athletes.

## Figures and Tables

**Figure 1 sensors-23-00058-f001:**
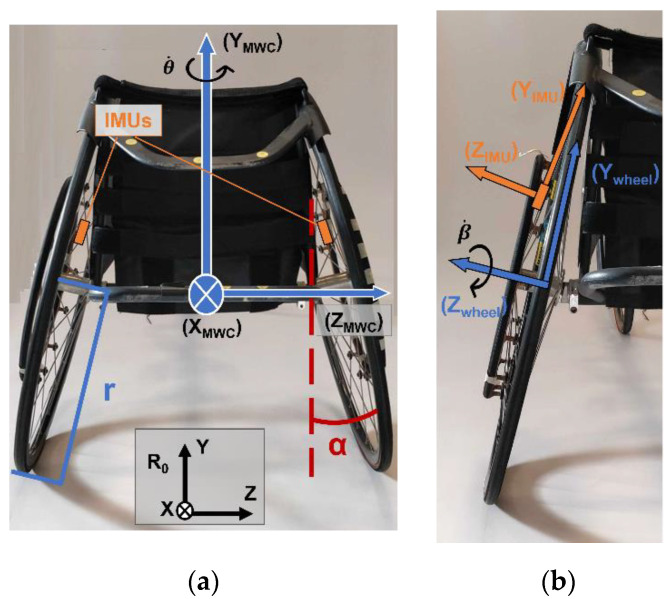
(**a**) Back view of a wheelchair illustrating the placement of the IMUs, wheelchair reference frame used as well as camber (α), and rear wheel radius (r); (**b**) Illustration of the tilt of an IMU placed on the spokes of a wheelchair.

**Figure 2 sensors-23-00058-f002:**
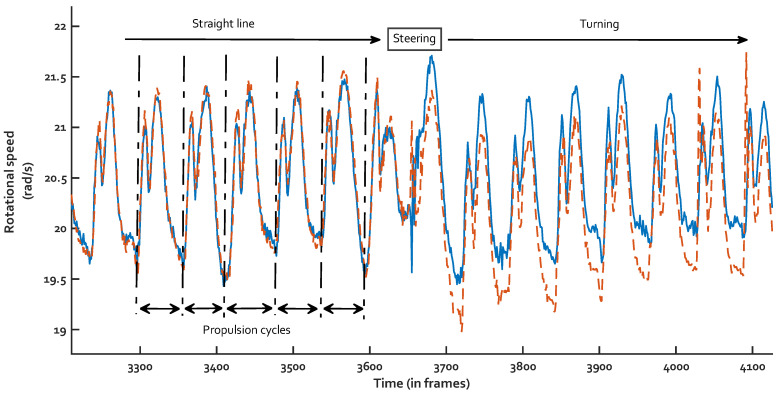
Cycle identification on MWC wheels rotational velocity curves. [Blue continuous line = right wheel; Orange dashed line = left wheel].

**Figure 3 sensors-23-00058-f003:**
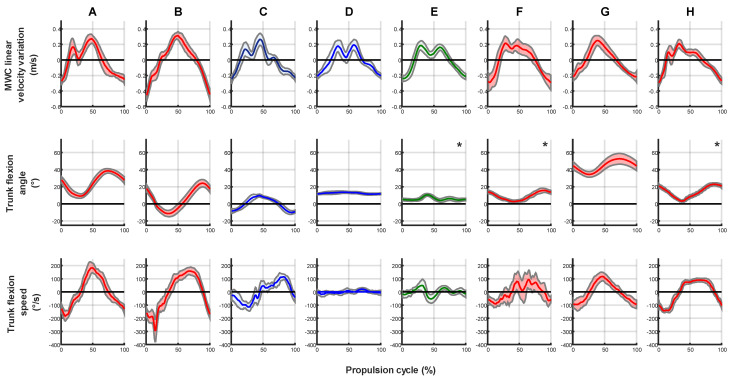
Average propulsion cycle of each athlete. First row: MWC linear velocity (centered on the average cycle speed); Second row: trunk flexion; Third row: trunk flexion speed. The average cycle of each athlete is available as Appendix A. [Red (T54), blue (T53) and green (T52) line = mean over all cycles identified per athlete; shaded corridor = mean ± 1 std; *: data with a potential offset due to the visual estimation of the initial inclination of the IMU placed on the trunk].

**Table 1 sensors-23-00058-t001:** Athletes’ demographics and wheelchair characteristics. The coordinates of the center of gravity are given in the MWC local frame defined by its origin placed at the center of the axis connecting the two rear wheel hubs, with the *x*-axis forward, the *y*-axis upward and the *z*-axis medial-lateral to the right (Figure 1a).

Athlete		A	B	C	D	E	F	G	H
Gender		M	M	M	M	M	M	M	M
Age		20	18	50	32	42	39	52	19
Impairment		sacral agenesis	Bilateral amputee	Poliomyelitis	SCI	SCI	SCI	Amputee	Amputee
Classification		T54	T54	T53	T53	T52	T54	T54	T54
Level		Elite	Elite	Elite	Elite	Elite	Elite	Intermediate	Elite
IMU used		Xsens	Xsens	Xsens	Xsens	WheelPerf	WheelPerf	Xsens	WheelPerf
Position of the trunk IMU		Sternum	Sternum	Sternum	Back	Back	Back	Sternum	Back
Mass (kg)		9.66	9.36	8.36	9.3	10.8	10.1	/	9.8
Camber (°)		13	12	11	11.5	12	12	11	12
Rear wheel diameter (cm)		69	71	71	66	71	71	66	68
Handrim diameter (cm)		40	38	34	38	35	/	37	/
Center of gravity (cm)	x	29.0	26.1	22.4	23.5	/	/	/	/
	y	38.1	41.1	41.0	42.4	/	/	/	/
	z	−0.1	−0.5	0.4	0.3	/	/	/	/
Yaw mass moment of inertia (kg.m^2^)		2.1	2	2.5	2.3	/	/	/	/
Wheelbase (m)		1.3	1.2	1.3	1.3	/	/	1.3	/

**Table 2 sensors-23-00058-t002:** Number of cycles studied per athlete, time to perform 400m, average propulsion parameters (cycle time, propulsion cycle speed and absolute linear velocity variation) and average trunk kinematics. Standard deviations are presented in parenthesis. *: data with a potential offset due to the visual estimation of the initial inclination of the IMU placed on the trunk.

Athlete	A	B	C	D	E	F	G	H
Number of cycles studied	28	50	12	28	60	32	37	28
400 m performance (s)	61.81	56.20	54.68	58.52	77.73	54.33	73.97	54.4
Average cycle time (s)	0.60 (0.04)	0.60 (0.02)	0.58 (0.05)	0.58 (0.05)	0.57 (0.04)	0.50 (0.04)	0.58 (0.04)	0.54 (0.04)
Propulsion cycle average speed (m/s)	7.04 (0.07)	7.95 (0.10)	8.55 (0.14)	6.99 (0.12)	5.80 (0.31)	8.56 (0.20)	5.61 (0.32)	8.52 (0.17)
Velocity variation amplitude (m/s)	0.59 (0.06)	0.83 (0.10)	0.58 (0.05)	0.48 (0.05)	0.49 (0.04)	0.60 (0.13)	0.56 (0.05)	0.54 (0.05)
Average trunk flexion (°)	25 (2.0)	5 (3.2)	0 (1.4)	13 (1.1)	6 (1.0) *	11 (1.1) *	44 (3.7)	14 (1.1) *
Average minimal trunk flexion (°)	39 (2.7)	25 (4.1)	10 (1.5)	14 (1.5)	11 (1.2) *	18 (1.6) *	54 (5.6)	24 (1.7) *
Average maximal trunk flexion (°)	9 (2.4)	−12 (3.6)	−11 (1.8)	11 (1.1)	2 (1.2) *	3 (2.2) *	33 (2.9)	2 (1.1) *
Average trunk flexion amplitude (°)	31 (2.4)	37 (4.5)	21 (1.5)	3 (1.3)	9 (1.7)	15 (2.5)	21 (4.9)	21 (1.8)
Average maximal trunk flexion speed (°/s)	211 (35)	180 (20)	133 (15)	33 (8)	91 (33)	146 (38)	140 (32)	103 (11)

**Table 3 sensors-23-00058-t003:** Pearson Correlation coefficients (R) between kinematic and spatiotemporal parameters monitored. Correlation coefficients greater than 0.9 in absolute value are presented in bold. [Class = Classification level; Time = time to perform 400 m; # peaks = number of peaks in velocity profile; CT = Cycle time; AV = Average velocity; VV = Velocity variation; ATF = Average trunk flexion; MinTF = Min trunk flexion; MaxTF = Max trunk flexion; TFA = Trunk flexion amplitude; TFS = Average trunk flexion speed].

Parameter	Class	Time	# Peaks	CT	AV	VV	ATF	MinTF	MaxTF	TFA	TFS
Class	**1.00**	−0.46	−0.62	−0.14	0.37	0.50	0.45	0.62	0.18	0.64	0.61
Time		**1.00**	−0.10	0.28	**−0.94**	−0.37	0.47	0.35	0.58	−0.27	−0.16
# Peaks			**1.00**	0.15	0.14	−0.51	−0.48	−0.59	−0.33	−0.39	−0.28
CT				**1.00**	−0.42	0.21	0.19	0.31	0.04	0.39	0.12
AV					**1.00**	0.37	−0.59	−0.45	−0.70	0.28	0.21
VV						**1.00**	−0.19	0.13	−0.48	0.82	0.65
ATF							**1.00**	**0.94**	**0.92**	0.11	0.17
MinTF								**1.00**	0.74	0.45	0.44
MaxTF									**1.00**	−0.28	−0.16
TFA										**1.00**	0.83
TFS											**1.00**

## Data Availability

The data presented in this study are available in supplementary material.

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
