# Peer review of "Analyzing Intra-Cycle Velocity Profile and Trunk Inclination during Wheelchair Racing Propulsion"

_sensors, 2022, doi:10.3390/s23010058_

Round 1

Reviewer 1 Report

The manuscript 'Analyzing intra-cycle velocity profile and trunk inclination during wheelchair racing propulsion' is well presented and with complete experimental verification in a very interesting subject. 

So, it would be advisable to publish it after the following minor advisors:

1. Please, change the first letter in capital for inertial measurement units (IMUs)

2. The label description of the table is too extended, add this description to the main text.

3. Please, clarify the scientific contribution of the study clearly

Reviewer 2 Report

This work is very interesting for me, because is based on the experimental data in very specific sport discipline.  

I gave an excellent estimations, but the paper has a following defects:

1. The number of participants is only 8 with strongly individual characteristics (diagnosis, level of diagnosys, probably the more than one diagnosis for older participants). This means that the authors can discuss the obtained results only as tendencies.

2. The  participant diagnosis are different. This probably means (I am not doctor and I am not competent in these diagnosis) that the coordination and muscle synergy between upper and lower extremity is completely different. This means that the Pearson Correlation coefficients obtained must be analysed very carrefully. 

Improvement:

2.1. The authors must collect the participants with a same diagnosis to improve the paper quality . 

2.2. Statistical analysis. The obtained signals are preprocessed by special ??? software (from two different manifacturers - Xsens, 100Hz or WheelPerf System, 128Hz)) which generates averaged time series for complete cyclic movement with time stamps of fractions in between [0,100]. Thus, the available averaged data are strongly individual, strongly periodic and non independent and identicaly distributed. Because, time stamps probably are not synchronized between participants with different IMUs, the data sets are aggregated by averaging to integer value timestamps, i.e. 0,1,2,…,100. Here in my opinion it must be apply Data model calibration before Pearson Correlation.

3. Why participant C has 12 numbers of cycles studied? For 400 m distance the cycles must be more. 

Improvement: If the participant C data were inaccurate, remove them. 

3. The obtained signals for cycle parameters are strictly individual and are preprocessed by special ??? software from two different manifacturers - Xsens, 100Hz or WheelPerf System, 128Hz. When the obtained results in the paper were statistically treated with Pearson Correlation, the authors lost the individual trends.   

Improvement: The authors can attempt to find individual statistical relations between the concrete participant obtained parameters data, using appropriate statistical test. 

Round 2

Reviewer 2 Report

No coment.